# Decorin Protects Cardiac Myocytes against Simulated Ischemia/Reperfusion Injury

**DOI:** 10.3390/molecules25153426

**Published:** 2020-07-28

**Authors:** Renáta Gáspár, Kamilla Gömöri, Bernadett Kiss, Ágnes Szántai, János Pálóczi, Zoltán V. Varga, Judit Pipis, Barnabás Váradi, Bence Ágg, Tamás Csont, Péter Ferdinandy, Monika Barteková, Anikó Görbe

**Affiliations:** 1Metabolic Diseases and Cell Signaling (MEDICS) Research Group, Department of Biochemistry, Interdisciplinary Excellence Centre, University of Szeged, Dom ter 9, H-6720 Szeged, Hungary; gaspar.renata@med.u-szeged.hu (R.G.); csont.tamas@med.u-szeged.hu (T.C.); 2Cardiovascular Research Group, Department of Pharmacology and Pharmacotherapy, University of Szeged, Dom ter 12, H-6720 Szeged, Hungary; gomori.kamilla@med.u-szeged.hu (K.G.); szantai.agnes@med.u-szeged.hu (Á.S.); paloczi.janos@med.u-szeged.hu (J.P.); 3Cardiometabolic Research Group, Department of Pharmacology and Pharmacotherapy, Semmelweis University, Nagyvarad ter 4, H-1089 Budapest, Hungary; kiss.bernadett@med.semmelweis-univ.hu (B.K.); varga.zoltan@med.semmelweis-univ.hu (Z.V.V.); barnabas36@gmail.com (B.V.); agg.bence@med.semmelweis-univ.hu (B.Á.); peter.ferdinandy@pharmahungary.com (P.F.); 4MTA-SE System Pharmacology Research Group, Department of Pharmacology and Pharmacotherapy, Semmelweis University, H-1089 Budapest, Hungary; 5Pharmahungary Group, Hajnoczy utca 6, H-6722 Szeged, Hungary; judit.pipis@pharmahungary.com; 6Institute for Heart Research, Centre of Experimental Medicine, Slovak Academy of Sciences, Dúbravská cesta 9, 841 04 Bratislava, Slovak; 7Institute of Physiology, Comenius University in Bratislava, Sasinkova 2, 813 72 Bratislava, Slovak

**Keywords:** cardiac myocytes, cardio protection, ischemia/reperfusion injury, proteoglycan, decorin

## Abstract

Search for new cardioprotective therapies is of great importance since no cardioprotective drugs are available on the market. In line with this need, several natural biomolecules have been extensively tested for their potential cardioprotective effects. Previously, we have shown that biglycan, a member of a diverse group of small leucine-rich proteoglycans, enhanced the expression of cardioprotective genes and decreased ischemia/reperfusion-induced cardiomyocyte death via a TLR-4 dependent mechanism. Therefore, in the present study we aimed to test whether decorin, a small leucine-rich proteoglycan closely related to biglycan, could exert cardiocytoprotection and to reveal possible downstream signaling pathways. Methods: Primary cardiomyocytes isolated from neonatal and adult rat hearts were treated with 0 (Vehicle), 1, 3, 10, 30 and 100 nM decorin as 20 h pretreatment and maintained throughout simulated ischemia and reperfusion (SI/R). In separate experiments, to test the mechanism of decorin-induced cardio protection, 3 nM decorin was applied in combination with inhibitors of known survival pathways, that is, the NOS inhibitor L-NAME, the PKG inhibitor KT-5823 and the TLR-4 inhibitor TAK-242, respectively. mRNA expression changes were measured after SI/R injury. Results: Cell viability of both neonatal and adult cardiomyocytes was significantly decreased due to SI/R injury. Decorin at 1, 3 and 10 nM concentrations significantly increased the survival of both neonatal and adult myocytes after SI/R. At 3nM (the most pronounced protective concentration), it had no effect on apoptotic rate of neonatal cardiac myocytes. No one of the inhibitors of survival pathways (L-NAME, KT-5823, TAK-242) influenced the cardiocytoprotective effect of decorin. MYND-type containing 19 (Zmynd19) and eukaryotic translation initiation factor 4E nuclear import factor 1 (Eif4enif1) were significantly upregulated due to the decorin treatment. In conclusion, this is the first demonstration that decorin exerts a direct cardiocytoprotective effect possibly independent of NO-cGMP-PKG and TLR-4 dependent survival signaling.

## 1. Introduction

There is an unmet clinical need for cardioprotective therapies against myocardial ischemia-reperfusion (I/R) injury [1,2]. Therefore, novel potential cardioprotective therapies are much sought for. In line with this need, a plenty of substances including natural biomolecules have been extensively tested for their cardioprotective potential. Proteoglycans are potential cardioprotective macromolecules of the extracellular matrix (ECM) which have been shown to play important roles in the ECM such as cell adhesion, deposition of collagens, activation of cytokines and growth factors or matrix assembly [3,4]. Previously, we have shown that exogenous administration of biglycan, which denotes a diverse group of small leucine-rich proteoglycan family, protects myocardial cells from simulated ischemia and reperfusion (SI/R) injury via Toll-like receptor-4-mediated mechanisms involving activation of survival kinases such as ERK, JNK and p38 MAP kinases and increased nitric oxide (NO•) production [5,6]. We also have shown that a number of cardioprotective genes are up-regulated in hearts of biglycan overexpressing mice [7].

In line with the cardioprotective effects of biglycan it is plausible to speculate that another proteoglycan, decorin, could also exert cardio protection. Decorin and biglycan as two members of a small leucine-rich proteoglycan family of ECM share many common features such as structural similarities or similar size of their core proteins (40 kD); however, there are also some differences making decorin potentially more favorable for application than biglycan. Decorin is usually “decorated” with only one chondroitin or dermatan sulfate chain while biglycan can be decorated with either one or two chains, thus decorin represents a less diverse group of molecules with significantly lower molecular weight than biglycan [8]. Moreover, production of these proteoglycans in the same cell line (HT 1080) yields 3 times higher amount of recombinant decorin (30 mg from 10^9^ cells per 24 h) [9] than biglycan (10 mg from 10^9^ cells per 24 h) [10].

Recent evidence indicates that decorin is included in a broad range of cellular processes including collagen fibrillogenesis, wound repair, angiostasis, tumor growth and autophagy [11]. In cardiovascular system it is proposed to play a key role in the proper tissue scar formation following myocardial infarction [12]. Decorin is a bi-functional proteoglycan, besides being a component of ECM, as a signaling molecule it interacts with different tyrosine kinase receptors including the epidermal growth factor receptor (EGFR), the insulin-like growth factor-1 receptor (IGF-IR) or the vascular endothelial growth factor receptor 2 (VEGFR2), as well as with the innate immunity receptors Toll-like receptors -2 and -4 (TLR-2, TLR-4), further leading to activation of p38, MAPK, NF-κB pathways and enhanced synthesis of the proinflammatory cytokines TNFα and IL-12 [13]. Recently, a protective effect of decorin on acute I/R injury in rat kidneys has been documented [14].

The role of decorin in cardiovascular diseases was recently reviewed by Vu et al., they summarized the role and function of decorin in cardiovascular diseases. Decorin interacting via cell surface receptor regulates plenty of cellular functions (reduces TGF-β dependent fibrosis, proliferation of myofibroblasts and endothelial cells, renal and neuronal protection from ischemic assaults) but these processes are dependent on cell types and pathological circumstances [15].

However, it is not known if decorin exerts acute cardiocytoprotective effect.

Therefore, the aim of our present study was to test if decorin exerts cardioprotective effects against simulated ischemia/reperfusion injury in primary cultures of isolated neonatal as well as adult rat cardiomyocytes and to reveal if well-known cellular survival signaling pathways involved in these effects.

## 2. Results

### 2.1. Concentration-Dependent Effect of Decorin on Cell Viabilityand Proliferation in Isolated Neonatal Rat Cardiac Myocytes (NRCMs) in Normoxic Conditions and Its Protective Effect after SI/R

Treatment of the Neonatal Rat Cardiac Myocytes (NRCMs) with different concentrations of decorin for 4 + 2 h in normoxic conditions showed that decorin applied in 3 nM and 10 nM concentrations significantly increased cell viability of NRCMs while 1 nM, 30 nM and 100 nM concentrations has not changed when compared to the vehicle-treated normoxic group (Figure 1a).

The effect of decorin on cell proliferation could distort the results of cell viability, therefore we checked it in isolated NRCMs under normal circumstances. The rate of cell proliferation was measured by BrdU incorporation into the cells. Decorin at 1 nM concentration significantly increased cell proliferation in NRCMs while other concentrations had no effect (Figure 1b). In a separate experiment we investigated the direct effect of decorin on NRCMs in an acute in vitro model of SI/R injury. Treatment with different concentrations of decorin significantly increased the cell viability of NRCMs after 4 h/2 h of SI/R in comparison to vehicle-treated group in a concentration-dependent manner, showing significant efficacy at 1 nM, 3 nM and 10 nM concentrations (Figure 2a).

### 2.2. Protective Effect of Decorin on Cell Viability in Isolated Adult Rat Cardiac Myocytes (ARCMs) Exposed to SI/R

To test the effect of decorin on the cell viability in isolated ARCMs, which are more sensitive to hypoxia, cells were exposed to 30 min/2 h of SI/R and the most efficacious three concentrations of decorin were used: 1 nM, 3 nM and 10 nM. There was a significant difference between Normoxia control group and simulated ischemia treated with only vehicle, which means that the SI/R protocol was successful. All concentrations of decorin attenuated SI/R-induced cell death as compared to the vehicle-treated group (Figure 2b).

### 2.3. The Effect of Different Inhibitors of Survival Pathways on Decorin-Induced Cardioprotection in NRCMs Exposed to SI/R

To explore potential involvement of NO• signaling in the mechanisms of cardioprotective effect of decorin against SI/R injury, the NO-synthase (NOS) inhibitor, L-NAME at 10 µM was used in combination with decorin in NRCMs exposed to SI/R. Decorin at 3 nM significantly increased cell viability but this effect was not affected by L-NAME co-treatment. However, cell viability was increased by L-NAME alone (Figure 3a).

After NO-synthase inhibition, we further investigated the potential involvement of NO/cGMP/PKG signaling pathway by using the protein kinase G (PKG) inhibitor KT-5823 at 60 nM. KT-5823 was used alone and combined with decorin in NRCMs exposed to SI/R. Decorin at 3 nM significantly increased cell viability and this effect was not affected by KT-5823 co-treatment. However, cell viability was significantly improved by KT-5823 alone (Figure 3b).

Similar results have been shown when the potential role of decorin triggered TLR-4 activation was investigated in SI/R injury. The TLR-4 inhibitor TAK-242 was used at 50 µM together with decorin in NRCMs exposed to SI/R. Decorin at 3 nM significantly increased cell viability and the protective effect was not affected by TAK-242 co-treatment. However, as it showed with the other inhibitors, cell viability was significantly improved by TAK-242 alone (Figure 3c).

### 2.4. The Effect of 3 nM Decorin on Apoptosis in Isolated NRCMs Exposed to SI/R

Having seen the most pronounced cardiocytoprotection at 3 nM of decorin treatment on cell viability of NRCMs exposed to SI/R, in a separate experiment we collected data on the rate of apoptosis. Treatment of the NRCMs with 3 nM concentration of decorin for 20 + 4 h in simulated ischemia/reperfusion showed no difference in apoptosis neither measured by TUNEL fluorescence staining (Figure 4a), nor by caspase-3/7 activity (Figure 4b) compared to the vehicle-treated group.

### 2.5. Akt and Activated p-Akt Protein Level in NRCMs

In the western blot measurement 20 + 4 h treatment with different concentrations of decorin (vehicle, 1, 3, 10, 30 or 100 nM) showed no difference between groups in case of total Akt/GAPDH ratio under normoxic conditions (Figure 5b). Although phosphorylated-Akt (p-Akt) shows concentration-dependent manner of protein expression in NRCMs, there is no significant differences between groups (Figure 5b). In separate experiment NRCMs went through 4 h of simulated ischemia and the increasing decorin concentration showed elevated trends in Akt/GAPDH ratio. On the contrary, a significant decrease of p-Akt shown in case of 3, 10, 30 nM of decorin. (Figure 5c). Phosphorylation of Akt does not show similarity either to Akt level or normoxic p-Akt/Akt ratio. For uncropped western blot images, see Appendix A (Appendix A).

### 2.6. Differentially Expressed mRNA from NRCM

After 20 h of 3nM decorin/vehicle treatment NRCMs were exposed to 4 h of simulated ischemia and mRNA expression changes were identified by sequencing from the cell lysate. By RNA sequencing out of the 29,496 genes annotated in the reference annotation 19,487 were detectable and 419 showed differential expression if no correction for multiple comparisons was applied. After correction for multiple comparisons 2 genes, namely zinc finger, MYND-type containing 19 (Zmynd19) and eukaryotic translation initiation factor 4E nuclear import factor 1 (Eif4enif1) were observed to be significantly upregulated due to the decorin treatment (Figure 6).

### 2.7. Gene Ontology Analysis

To explore biological processes modified by decorin treatment, Gene Ontology (GO) enrichment analysis was performed. The result of the GO analysis clearly showed that differentially expressed mRNAs were significantly associated with for example, response to oxidative stress, response to antibiotic, mitotic cell cycle, cellular macromolecule metabolic process, organonitrogen compound metabolic process and nitrogen compound metabolic process (Table 1).

## 3. Discussion

We have shown here that decorin exerted cardiocytoprotective effect in both isolated neonatal and adult rat cardiomyocytes exposed to SI/R. The effect of decorin was not influenced by inhibitors of well-known survival signaling pathways, that is, the NO-synthase inhibitor, L-NAME, the PKG inhibitor, KT-5823 or the inhibitor of TLR-4, TAK-242, respectively. This is the first demonstration of the cardiocytoprotective effect of decorin and its action on gene expression levels in cardiac myocytes; however, the mechanism of its action still remains unclear.

Decorin has been shown to be involved in the ventricular remodeling following acute myocardial infarction [16,17,18] and to be required for the proper fibrotic evolution of myocardial infarctions [12]. Recent studies have demonstrated that post-infarction gene therapy with adenoviral vector expressing decorin mitigates cardiac remodeling and dysfunction [19] suggesting promising therapeutic potential of exogenous decorin for the treatment of acute myocardial infarction. Gubbiotti et al. showed in decorin knock out (Dcn-/-) and wild type mice that decorin has a new role as a nutrient sensor that modulates cardiac autophagy and metabolism [20] and exogenous decorin treatment (10 mg/kg core protein) restores fasting-induced autophagy in Dcn-/- hearts and can also salvage cardiac function after 25 h of fasting [21].

Our study aimed to show the direct potential cardiocytoprotective effect of decorin in isolated cardiomyocytes in an acute in vitro model of simulated ischemia/reperfusion injury. The findings of our present study documents direct cardiocytoprotective effect of decorin in ARCMs as well as NRCMs exposed to SI/R. Similarly, as findings of our previous study documenting a dose-dependent protective effect of another matrix proteoglycan – biglycan [6], the cardioprotective effect of decorin on NRCMs cell viability exposed to SI/R was dose-dependent. The protective effect of decorin on ARCMs in the present study has been shown to be dose-independent but it should be pointed out that for ARCMs, only those concentration of decorin were used which exerted protective effects in NRCMs. Taken together, small extracellular matrix proteoglycans including biglycan and decorin may represent a powerful tool for cardio protection. Biglycan and decorin have been shown to differentially regulate signaling in the fetal membranes suggesting their different biological activities [22]. There are also differences in enzymatic degradation of these two proteoglycans by matrix metalloproteinases (MMPs) [23,24] which points to different regulation of these two proteoglycans and may also potentially influence their biological activities. Our present study suggests that the molecular mechanisms involved in the cardioprotective effect of decorin are different from those of biglycan. While cardiocytoprotective action of biglycan includes activation of TLR-4 and its downstream effectors [6], the protective effect of decorin documented in the present study seems to be TLR-4 independent. This might be explained by possibly different cellular effects of decorin and biglycan via TLR signaling found for example in tumor cells, where opposite effects of these proteoglycans on tumor growth via TLR have been shown [25,26]. In addition, the protective effect of biglycan has been shown to involve enhanced production of NO•, while the effect of decorin in the present study was not affected by NOS inhibition suggesting a NO-independent action of decorin in preventing the SI/R-induced cell death of NRCMs. Finally, inhibition of PKG in the current study has also no impact on decorin action in NRCMs exposed to SI/R suggesting these effects to be independent of NO-cGMP-PKG signaling. It could be concluded that molecular mechanisms of biglycan and decorin cardiocytoprotective action in I/R injury are different. Gaspar et al. showed that pharmacological blockade of Toll-like receptor 4 (TLR-4) signaling and its downstream signaling contributors (IRAK1/4, ERK, JNK and p38 MAP kinases) abolished the cytoprotective effect of exogenously administered recombinant biglycan core protein against SI/R injury [6]. In another study, decorin secreted by human adult renal stem cells through the TLR-2 receptor induce renal tubular cell regeneration [27].

This can be explained by structural differences [8] as well as by differences in their biological actions which have been documented in some studies [22]. Besides cardiac myocytes, other myocardial cell types, like cardiac fibroblasts, may also contribute to cardio protection. Expression of proteoglycans by fibroblasts is influenced by TNF-α and TGF-β [28]. Recently it was documented that certain amount of decorin is produced in the scar tissue, the border zone close to the infarction, as well as in remote region of pig hearts after myocardial infarction. In this study, biglycan and decorin levels were higher in the border zone and are suggested to support the stabilization of collagen fibrillogenesis [29].

Present study showed that exogenously administered decorin improved viability and triggered proliferation rate of NRCMs even in normoxic condition. Viability assay based on measurement of intracellular enzyme activity, meanwhile proliferation assay based on cell division, how much tagged uracil (BrdU) is able to incorporate in. The results of the present study indicate that decorin at 1nM concentration increased while at 100 nM decreased cell proliferation of NRCMs. Therefore, the positive effect of 1 nM decorin seen in the viability assay after SI/R can be due to the proliferative effect of this concentration of decorin. However, decorin at other concentrations did not affect proliferation, therefore, the results of viability assay of these concentrations of decorin have not been influenced by proliferation. The ability of decorin to differentially regulate the cell proliferation is well known for many years [30] and the anti-proliferative effects of decorin are considered to be involved in its anti-cancer effects [31,32]. On the other hand, decorin has been shown to promote the proliferation of myoblasts [33,34] and that gene transfer of decorin in vivo promotes mice skeletal muscle regeneration and accelerates muscle healing after injury [35]. Moreover, the antimyostatin effect of decorin cleavage products [36] may be also involved in the decorin-induced proliferation of cardiac myocytes seen in the present study.

Decorin affects rate of apoptosis in different experimental models. Decorin induced apoptosis via activation of caspase-3 in A431 tumor engraftment cells [37]. In another model using skin fibroblasts decorin treatment resulted a significant increase in the expression of apoptotic markers, histone-1, caspase-1, caspase-8 and p53 in superficial fibroblasts when compared with deep dermal fibroblasts [38]. Overexpression of decorin inhibited mesangial cells proliferation by inducing apoptosis and cell growth arrest in vitro and it also downregulates expression of TGF-beta1 [39]. Therefore we evaluated the apoptotic effect of decorin in simulated ischemic conditions by measuring caspase-3 activity and by performing TUNEL assay in NRCMs. In the present study, however, decorin treatment had no significant effect on apoptosis in neonatal cardiac myocytes.

P-Akt/Akt is considered as pro-survival protein which is included in the downstream signaling pathway for example reperfusion injury salvage kinase (RISK) pathway [40], therefore Akt kinase has a cardioprotective role against ischemia/reperfusion injury. We evaluated the activity of Akt kinase by measuring the p-Akt/Akt ratio (Akt is activated by its phosphorylation) by western blot to see if an Akt-dependent pathway is potentially involved in the mechanism of decorin action.

Suzuki et al. investigated the effect of decorin in myogenic cells. Decorin activated Akt downstream of IGF-IR and enhanced the differentiation of C2C12 myoblast cells [41]. In endothelial cells decorin inhibited anti-autophagic signaling via suppression of Akt/mTOR/p70S6K activity with the concurrent activation of pro-autophagic AMPK-mediated signaling cascades [42]. Activation of autophagy in vitro using mouse embryonic NIH-3T3 fibroblasts induced decorin expression via mTOR pathway, while biglycan expression showed no changes [20]. Overexpression of decorin ameliorated diabetic cardiomyopathy and promoted angiogenesis through the IGF1R-Akt-VEGF signaling pathway in endothelial cells in vivo and in vitro [43]. To our best knowledge, there are no available data of the direct effect of decorin on Akt signaling in cardiac myocytes. Our results point to a decreased phosphorylation/activation of Akt after simulated ischemia, which is not in line with majority of data regarding to cardio protection against ischemia/reperfusion injury. Therefore, decorin probably protects cardiac cells activating different downstream signaling pathways.

We performed RNA sequencing analysis to monitor the direct effect of Decorin on isolated cardiac myocytes. This is the first demonstration that zinc finger, MYND-type containing 19/Zmynd19 and eukaryotic translation initiation factor 4E nuclear import factor 1(eIF4E) are significantly upregulated due to Decorin action. eIF4E is best known for its function in the initiation of protein synthesis on capped mRNAs in the cytoplasm and involved in the nuclear export of specific mRNAs [44]. It has been shown, that eIF4E plays an important role in the nucleocytoplasmic export of human iNOS mRNA in colon carcinoma cell line [45]. Decorin administration influenced NO• mediated pathway in our study as well, as using nitric oxide (NOS) inhibitor and downstream acting PKG inhibitor we found protective effect against ischemia/reperfusion induced injury. Culjkovic B. et al., showed that eIF4E associates and promotes the nuclear export of cyclin D1 in the nucleus and is involved in the regulation of cell proliferation [46]. In the present study, we showed that Decorin treatment in 1 nM concentration enhanced the proliferation of cardiac cell culture. This finding also in line with our findings using GO analysis, which presented overrepresentation mitotic cell cycle (GO:0000278) genes. GO analysis resulted overrepresentation genes in response to oxidative stress (GO:0006979) as well, which phenomenon has been studied in traumatic brain injury model, where Decorin protected neuronal cells reducing level of oxidative stress [47].

Limitations of the study—Although the present study clearly showed the cardiocytoprotective effect of decorin in both neonatal and adult cardiomyocytes, it did not reveal the mechanism of its action because none of the used inhibitors affected decorin-induced cardiocytoprotection. In our present study we used the same doses of inhibitors as in our previous studies [6,48] where they did not affect cell viability after SI/R on their own, while in the present study they affected cell viability. Although NOS, TLR-4 and PKG inhibitors did not affect decorin induced cytoprotection in the present study, the variability of response of cardiac myocytes to NOS, TLR-4 and PKG inhibitors in different studies may cause uncertainty as to whether the selected pathways may contribute to the cardiocytoprotective effects of decorin in the present study. Indeed, other studies also showed diverse effects of these inhibitors in cardio protection. For example PKG inhibitor KT-5823 in the same dose (1 µM) blocked the cardioprotective effect of ischemic postconditioning (IPostC) in some studies [49,50] while it had no effect on the cardioprotective effect of IPostC in other studies [51,52] in an isolated perfused heart. Also, opposite effects of TLR-4 inhibitor TAK-242 have been documented in different types of cardiomyocyte injury—it has been shown to dramatically block the high glucose-induced cytotoxicity leading to an increase in cell viability in H9c2 cells [53], on the other hand it attenuated biglycan-induced cardiocytoprotection and had no effect when applied alone in SI/R model in neonatal rat cardiomyocytes [6]. It has been shown that NOS inhibitor L-NAME at the dose of 30 µM exerted both cardioprotective [54] as well as no effects [55] in isolated hearts exposed to I/R. It is well known that the different doses of NOS inhibitors can induce either cardio protection or block cardioprotective pathways, see for reviews References [56,57]. Unfortunately, there is no study showing the dose response curve for TAK-242 and KT-5823 in I/R injury. This limitation reflects the fact that multiple pathways may be individually and sequentially activated in cardiomyocytes due to I/R [58,59,60] which may cause the variability of cardiomyocyte responses to I/R as well as to NOS, TLR-4, PKG inhibitors used in individual experiments. Indeed, a recent study by Heusch’s group [61] showed that the reproducibility of studies on cell signaling of cardio protection, even performed in the same laboratory, can be variable.

## 4. Materials and Methods

### 4.1. Culturing Primary Neonatal Rat Cardiac Myocytes (NRCMs)

NRCMs were isolated from newborn Wistar rats as described previously [48]. Briefly, neonatal rats were sacrificed by cervical dislocation. The hearts were rapidly removed and placed in a cold phosphate buffered saline solution. After separation of atria, ventricles were minced with a fine forceps and collected in 0.25% trypsin (Gibco BRL, Rockford, IL, USA). Tissue fragments were further digested by trypsin for 25 min in a Falcon tube in 37 °C water bath. Then the cell suspension was centrifuged (450 × g for 15 min at 4 °C). The cell pellet was resuspended in culture medium-Dulbecco’s modified Eagle’s medium (DMEM) supplemented with 10% fetal bovine serum (FBS), L-Glutamine and AB/AM (Sigma-Aldrich, St. Louis, MO, USA). The single cell suspension was pre-plated in 6-well plates at 37 °C for 90 min to enrich the culture with cardiomyocytes. No cytostatic compound was added. The non-adherent myocytes were collected and plated at a density of 10^5^ cells/well onto 24-wells plates or 1.8 × 10^4^ cells/well in case of 96-wells plates. Culture medium was changed the day after preparation to 1% FBS containing differentiation medium. The cells were maintained at 37 °C in a standard CO_2_ incubator (Humidified atmosphere of 5% CO_2_).

### 4.2. Culturing Adult Rat Cardiac Myocytes (ARCMs)

ARCMs were isolated as described previously [6]. The hearts of 200–250 g male Wistar rats were excised after euthasol anesthesia (50 mg/kg) and heparin injection (50 U/kg). The hearts were then stabilized by retrograde aortic perfusion on a Langendorff system with solution A (In mM: NaHCO_3_ 25, KCl 4.7, NaCl 118.5, MgSO_4_-7H_2_O 1.2, KH_2_PO_4_ 1.2, glucose 10, Dyacetyl monoxime (BDM) 10 and CaCl_2_ 5 μM). Collagenase Type II (8000 U) was used for gentle digestion of tissue in solution B (in mM: NaHCO_3_ 25, KCl 4.7, NaCl 118.5, MgSO_4_-7 H_2_O 1.2, KH_2_PO_4_ 1.2, glucose 10, BDM 10, CaCl_2_ 50 μM, 1% BSA) for 30–45 min. The heart was further washed and minced in solution C (in mM: NaHCO_3_ 25, KCl 4.7, NaCl 118.5, MgSO_4_-7 H_2_O 1.2, KH_2_PO_4_ 1.2, glucose 10, CaCl_2_ 50 μM) and filtered. After filtration, the cells were washed 3 times with solution C. The calcium concentration was increased gradually to 1.8 mM. Cells were harvested in M199 medium (5% FBS, L-carnitine 5, taurine 5 and creatine-monohydrate 5mM). Cardiac myocytes were plated onto laminin coated (10 μg/mL) coverslips placed in 24-well plates at density of 7.5 × 10^3^ cell/well. After 3 h, the FBS-containing M199 was replaced with serum free M199. Two-day-old cultures were used for simulated ischemia/reperfusion experiments. Chemicals were purchased from Sigma-Aldrich (St. Louis, MO, USA).

### 4.3. Experimental Groups

NRCMs at day-2 were pretreated with 0 (Vehicle), 1, 3, 10, 30, 100 nM concentration of decorin for 20 h, respectively. Then 4 h SI and 2 h R or normoxic control treatment was applied as described above. Normoxic controls were treated with vehicle or decorin throughout the entire investigation (20 h, 4 h, 2 h). At the end of the reperfusion, the cell viability was measured with calcein assay. ARCMs were treated with 1, 3, 10 nM concentrations only during SI/R.

In separate experiments, we investigated the mechanism of action of decorin. For assessment of the role of NO• in the protective effect of decorin, the cells were treated with 3 nM decorin in the absence or presence of 10 µM NO-synthase inhibitor L-nitro-arginine methyl ester (L-NAME, Sigma-Aldrich, Saint Louis, MO, USA) during 4 h SI [6]. Similar experimental design was performed to assess the role of protein kinase G (PKG) downstream to NO-cGMP with application of 600 nM KT-5823 (Selective PKG inhibitor) [62] and the role of TLR-4 signaling with application of 50 µM TAK-242 (TLR-4 inhibitor) [6].

### 4.4. Simulated Ischemia/Reperfusion (SI/R)

To simulate ischemic conditions, the culture medium was replaced with a hypoxic solution containing in mM: NaCl 119, KCl 5.4, MgSO_4_ 1.3, NaH_2_PO_4_ 1.2, HEPES 5, MgCl_2_ 0.5, CaCl_2_ 0.9, Na-lactate 20, BSA 0.1% pH 6.4. To induce hypoxia the cells were then placed in a tri-gas incubator gassed through with a mixture of 95% N_2_ and 5% CO_2_ for 240 min at 37 °C. Normoxic control cells were covered with normoxic solution containing in mM: NaCl 125, KCl 5.4, NaH_2_PO_4_ 1.2, MgCl_2_ 0.5, HEPES 20, MgSO_4,_ 1.3, CaCl_2_ 1, glucose 15, taurine 5, creatine-monohydrate 2.5 and BSA 0.1%, pH 7.4 and cells were kept in normoxic incubator. After simulated ischemia or normoxia, the cells were reoxygenated and the hypoxic medium was replaced by culture medium (Simulated reperfusion). NRCMs were subjected to SI for 4 h and 2 h R, while ARCMs were subjected to 0,5 h SI and 2 h R (Figure 7.). Previously, we have shown that the experimental model of isolated cultures of heart-derived cells exposed to SI/R represents a validated tool for testing potential cardioprotective effects of different substances [48,62,63].

### 4.5. Cell Viability Assay

Cell viability was assessed by a calcein assay performed in each group after 2 h reperfusion. The cell-permeant calcein-AM dye (PromoKine, Heidelberg, Germany) stains living cells to be converted to green-fluorescent calcein by intracellular non-specific esterases [64]. The growth medium was removed, then cells were washed with PBS twice and incubated with calcein (1 μM) dissolved in DMSO and further diluted in D-PBS, for 30 min in a dark chamber. Then the calcein solution was replaced with fresh PBS and the fluorescence intensity of each well was detected by fluorescent plate reader (FluoStar Optima, BMG Labtech). Fluorescence intensity was measured in well scanning mode (scan matrix: 10 × 10; scan diameter: 10 mm; bottom optic; no of flashes/scan point: 3; temp: 37 °C; excitation wavelength: 490 nm; emission wavelength: 520 nm).

In the case of ARCMs the cell number varied from well-to-well, therefore living cell number was expressed in ratio of total cell count. ARCMs were incubated with propidium iodide (PI, 50μM) for 7 min included digitonin (4–10 M) (Sigma-Aldrich, St. Louis, MO, USA) to permeabilize and kill the cells. Then the PI solution was replaced with fresh PBS and fluorescence intensity of each well was detected; scan matrix: 10 × 10; scan diameter: 10mm; bottom optic; no of flashes/scan point: 3; temp: 37 °C; excitation wavelength: 544 nm; emission wavelength: 610 nm. The cytoprotective effect of different compounds was compared to simulated ischemic control groups [65].

### 4.6. Proliferation Assay

Testing the proliferative effect of decorin, BrdU (5′-bromo-2′-deoxiuridine, Abcam, Cambridge, UK) incorporation assay was performed using the Cell proliferation ELISA kit. The cells were pretreated for 24 h with the BrdU-labeling solution (Final concentration 10 μM) followed by fixation and denaturation by 30 min incubation with FixDenat solution. Then, horseradish peroxidase conjugated anti-BrdU antibody was added to the wells and incubated for 90 min at room temperature. Finally, tetramethylbenzidine substrate was added for 30 min and the absorbance was measured using plate reader (FluoStar Optima, BMG Labtech) at 450/620 nm.

### 4.7. TUNEL Assay

DeadEnd Fluorometric TUNEL assay (Promega, G3250, Wisconsin, WI, USA) measures nuclear DNA fragmentation by catalytically incorporating fluorescein-12-dUTP. Detection of fragmented DNA in apoptotic NRCMs after SI/R was performed according to manufacturer instruction. Briefly, cells were fixed 4% methanol-free formaldehyde solution in PBS (pH 7.4) for 25 min, then washed and cells were permeabilized by 0.2% Triton^®^ X-100 solution in PBS for 5 min. After removing liquids, enzyme and fluorescent nucleotide mix were added to cells and incubated for 60 min at 37 °C. After washing steps cells were stained with DAPI (4′,6-diamidino-2-phenylindole, blue-fluorescent DNA stain). Green and blue fluorescence intensity of each well was detected by fluorescent plate reader (FluoStar Optima, BMG Labtech). Fluorescence intensity was measured in well scanning mode (scan matrix: 10 × 10; scan diameter: 10 mm; bottom optic; no of flashes/scan point: 3; temperature: 37 °C) excitation wavelength for TUNEL: 490 nm; emission wavelength: 530 nm and for DAPI excitation wavelength: 435 nm, emission wavelength: 460 nm.

### 4.8. Caspase Assay

Investigation of apoptosis was assessed by CellEvent Caspase-3/7 assay (Thermofisher Scientific, Rockford, IL, USA) performed on NRCMs after SI/R, according to the manufacturer’s instructions. The growth medium was removed, then cells were washed with PBS twice and incubated with caspase-3/7 reagent (1 μM) dissolved in DMSO and further diluted in D-PBS, for 30 min in a dark chamber. The Green Detection Reagent is a four-amino acid peptide (DEVD) conjugated to a nucleic acid-binding dye, which is non-fluorescent until cleaved by caspase-3/7. After activation of caspase-3/7 in apoptotic cells, the cleaved DEVD peptide is enabling the dye to bind to DNA and produce a bright green fluorogenic response. Then the caspase-3/7 reagent solution was replaced with fresh PBS and the fluorescence intensity of each well was detected by fluorescent plate reader (FluoStar Optima, BMG Labtech). Fluorescence intensity was measured in well scanning mode (scan matrix: 10 × 10; scan diameter: 10 mm; bottom optic; no of flashes/scan point: 3; temp: 37 °C; excitation wavelength: 503 nm; emission wavelength: 530 nm).

### 4.9. Western Blot Sample Collection

For western blot analysis cell lysate sample was collected from neonatal rat cardiomyocyte culture. NRCMs were plated onto 6 well plates at 5 × 10^5^ cell/well density. NRCMs at day-2 were pretreated with 0 (Vehicle), 1, 3, 10, 30, 100 nM concentration of decorin for 20 h, respectively. Then 4 h SI treatment was applied as described above, supplemented with decorin. Normoxic controls were treated with vehicle or 1, 3, 10, 30, 100 nM decorin throughout the entire investigation (20 h + 4 h). At the end of 24 h treatment with or without simulated ischemia, cells were washed twice with ice-cold D-PBS, scraped from wells and collected in homogenization buffer (1× Radioimmunoprecipitation assay buffer (RIPA) containing protease inhibitor cocktail and phosphatase inhibitors) collected in Eppendorf tubes. The number of cells for *n* = 1 (biological sample) was collected and pooled together from 2 wells. To reach *n* = 4 we collected cells from 4 independent isolation (4 separate weeks). Cells were sonicated with an ultrasonic homogenizer (10 s, 4 °C). The homogenate was centrifuged (14000 × g, 10 min, 4 °C), the supernatants were further concentrated using Amicon^®^ Ultra-4 Centrifugal Filter Units (10 kDa cut-off limit). Concentrated lysate were kept on −70°C until western blot measurement [6].

### 4.10. Western Blot

Protein concentration of the homogenates was determined with BCA Protein Assay Kit (Thermofisher Scientific, Rockford, IL, USA). 40 μg of protein was loaded on 8 or 10% polyacrylamide gels and was separated by standard SDS-PAGE followed by transfer of proteins onto PVDF membranes (90 V, 15 min, 110 V, 85 min), transfer overnight, 200mA. After the transfer, the membranes were checked with Ponceau solution (0.05g Ponceau powder in 5% acetic acid). Membranes were blocked (2 h, RT) in 0.05% Tris buffered saline (TBS)-Tween20 containing 5% non-fat milk. To detect intracellular proteins in neonatal cardiac myocytes treated with exogenous decorin, membranes were incubated with p-Akt (Cell Signaling, Danvers, MA, USA, #9271, 60 kDa), Akt (Cell Signaling, #92972, 60 kDa) (1:500) and GAPDH (Cell Signaling, #5174, 37 kDa) (1:10000) primary antibody overnight at 4 °C in 5% milk followed by incubation with anti-rabbit-HRP secondary antibody (1:2000, in case of GAPDH 1:10000) for 2 h, at room temperature. After washing, membranes were developed with an enhanced chemiluminescence kit. Bands were quantified by Image Lab (BioRad, Hercules, CA, USA, version 4.1.0.2177) software.

### 4.11. RNA Isolation

Independent experiments NRCMs were treated 20 h with 3 nM decorin or vehicle prior to 4 h of simulated ischemia. Cells were collected and lysed in 1 mL of QIAzol Lysis Reagent. (QIAgen). Then Total RNA was extracted using Direct-zol™ RNA MiniPrep System (Zymo Research) according to the manufacturer’s protocol. The RNA Integrity Numbers and RNA concentration were determined by RNA ScreenTape system with 2200 Tapestation (Agilent Technologies, Santa Clara, CA, USA) and RNA HS Assay Kit with Qubit 3.0 Fluorometer (Thermo Fisher Scientific, Waltham, MA, USA), respectively.

### 4.12. RNA Library Construction

For Gene Expression Profiling (GEx) library construction, QuantSeq 3‘ mRNA-Seq Library Prep Kit FWD for Illumina (Lexogen GmbH, Wien, Austria) was applied according to the manufacturer’s protocol. The quality and quantity of the library was determined by using High Sensitivity DNA1000 ScreenTape system with 2200 Tapestation (Agilent Technologies, Santa Clara, CA, USA) and dsDNA HS Assay Kit with Qubit 3.0 Fluorometer (Thermo Fisher Scientific, Waltham, MA, USA), respectively. Pooled libraries were diluted to 1.8 pM for 1 × 86 bp single-end sequencing with 75-cycle High Output v2 Kit on the NextSeq 550 Sequencing System (Illumina, San Diego, CA, USA) according to the manufacturer’s protocol.

For small RNA library construction, NEBNext Multiplex Small RNA Library Prep Set for Illumina (New England Biolabs, Ipswich, MA, USA) was applied according to the manufacturer’s protocol. The quality and quantity of the library QC was performed by using High Sensitivity DNA1000 ScreenTape system with 2200 Tapestation (Agilent Technologies, Santa Clara, CA, USA) and dsDNA HS Assay Kit with Qubit 3.0 Fluorometer (Thermo Fisher Scientific, Waltham, MA, USA), respectively. Pooled libraries were diluted to 1.8 pM for 2 × 43 bp paired-end sequencing with 75-cycle High Output v2 Kit on the NextSeq 550 Sequencing System (Illumina, San Diego, CA, USA) at the Xenovea Ltd. according to the manufacturer’s instructions. Raw and processed RNA-sequencing datasets were deposited in the ArrayExpress database (https://www.ebi.ac.uk/arrayexpress/) under the accession number of E-MTAB-9325.

### 4.13. Bioinformatics Evaluation of the RNA Sequencing Data

Adapter trimming, quality and length filtering of raw RNA sequencing reads were performed by Cutadapt (version 1.15) [66]. During quality and length filtering, reads with an average Phred quality score below 30 or a length less than 19 nt were excluded from further analysis. FastQC (version v0.11.8) and MultiQC (version v1.7) were used for quality control analysis [67]. Reads obtained this way were aligned to Rnor_6.0 NCBI Rattus norvegicus reference genome and were annotated using the corresponding reference annotation by HISAT2 (version 2.0.4) and featureCounts (version of Subread v2.0.0), respectively [68,69]. DESeq2 Bioconductor package was utilized for normalization and differential expression analysis [70]. Correction of the p-values for multiple comparisons was done by calculating the false discovery rate according (FDR) to Benjamini and Hochberg [71].

### 4.14. Gene Ontology Enrichment Analysis

The online PANTHER Overrepresentation Test (geneontology.org, version released on 7 April 2020 [72]) was performed against the Rattus norvegicus reference gene list to assess Gene Ontology (database version released on 23 March 2020) biological process terms enriched among genes that were differentially expressed when considering the non-corrected p-values. For the enrichment analysis Fisher’s exact test was applied with false discovery rate correction for multiple comparisons.

## 5. Conclusions

In conclusion, the small leucine-rich proteoglycan decorin exerts cardiocytoprotective effects against SI/R, suggesting a therapeutic potential of exogenously administered decorin for the treatment of acute myocardial infarction. The molecular mechanism of its action still remains to be uncovered; however, it seems to be independent of NO-cGMP-PKG and TLR-4 signaling.

## Figures and Tables

**Figure 1 molecules-25-03426-f001:**
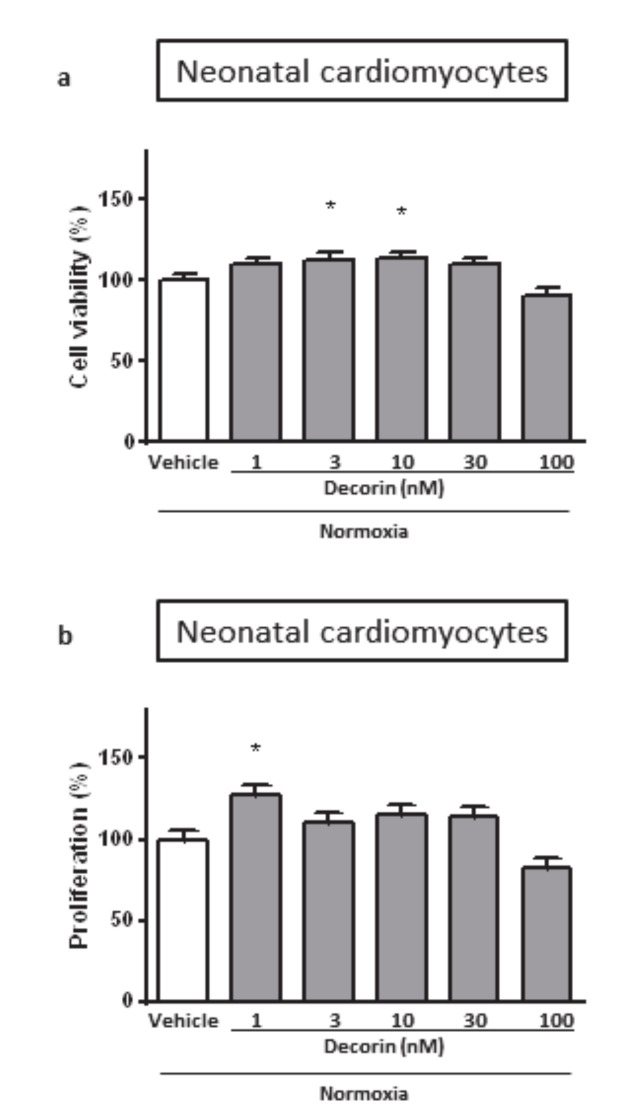
The effect of different concentrations of decorin on the (**a**) cell viability and (**b**) cell proliferation in Neonatal Rat Cardiac Myocytes (NRCMs) in normoxic conditions. Cell viability measured by calcein staining based on intracellular esterase’s enzyme activity. The rate of proliferation was evaluated by BrdU incorporation into the cells. Data are normalized to vehicle-treated Normoxia and presented as mean ± S.E.M. One-Way ANOVA, Dunnett’s multiple comparison test, **p* < 0.05 vs. vehicle treated cells (*n* = 5–6).

**Figure 2 molecules-25-03426-f002:**
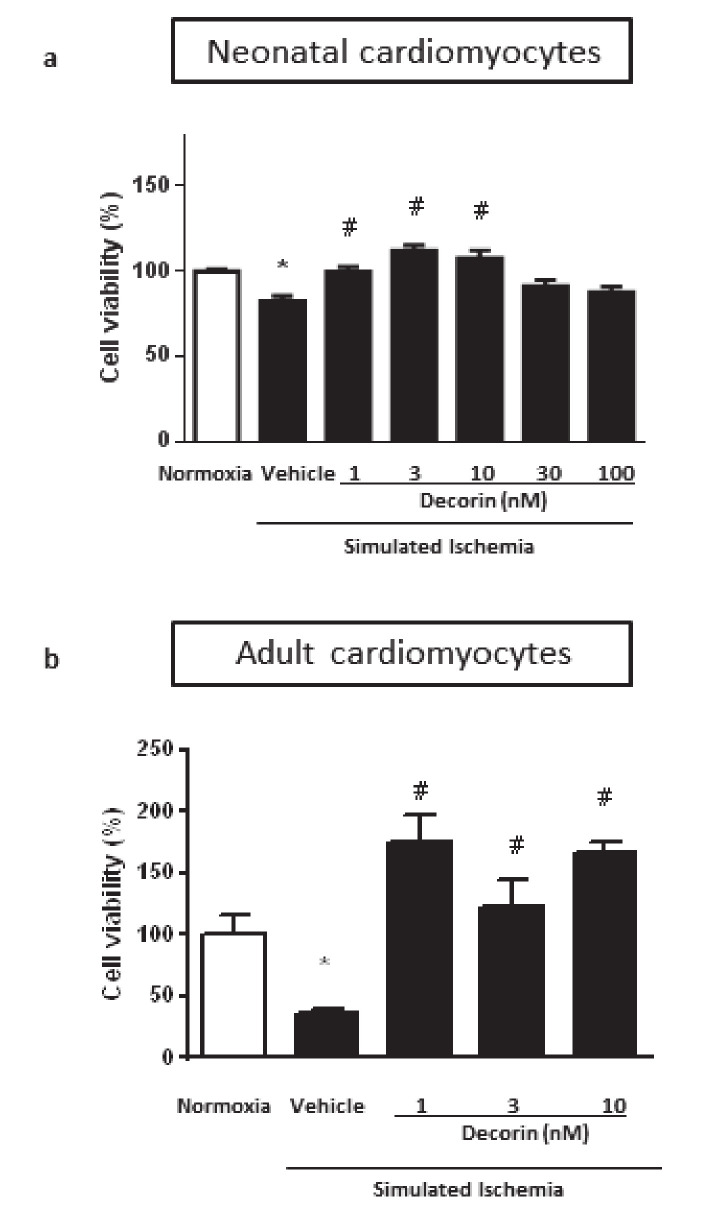
The effect of different concentrations of decorin on cell viability in isolated (**a**) NRCMs and (**b**) Adult Rat Cardiac Myocytes (ARCMs) exposed to SI/R. Data are normalized to vehicle-treated Normoxia and presented as mean ± S.E.M. One-Way ANOVA, Dunnett’s multiple comparison test, **p* < 0.05 vs. Normoxia and ^#^*p* < 0.05 vs. SI/R Vehicle treated cells (*n* = 5–6).

**Figure 3 molecules-25-03426-f003:**
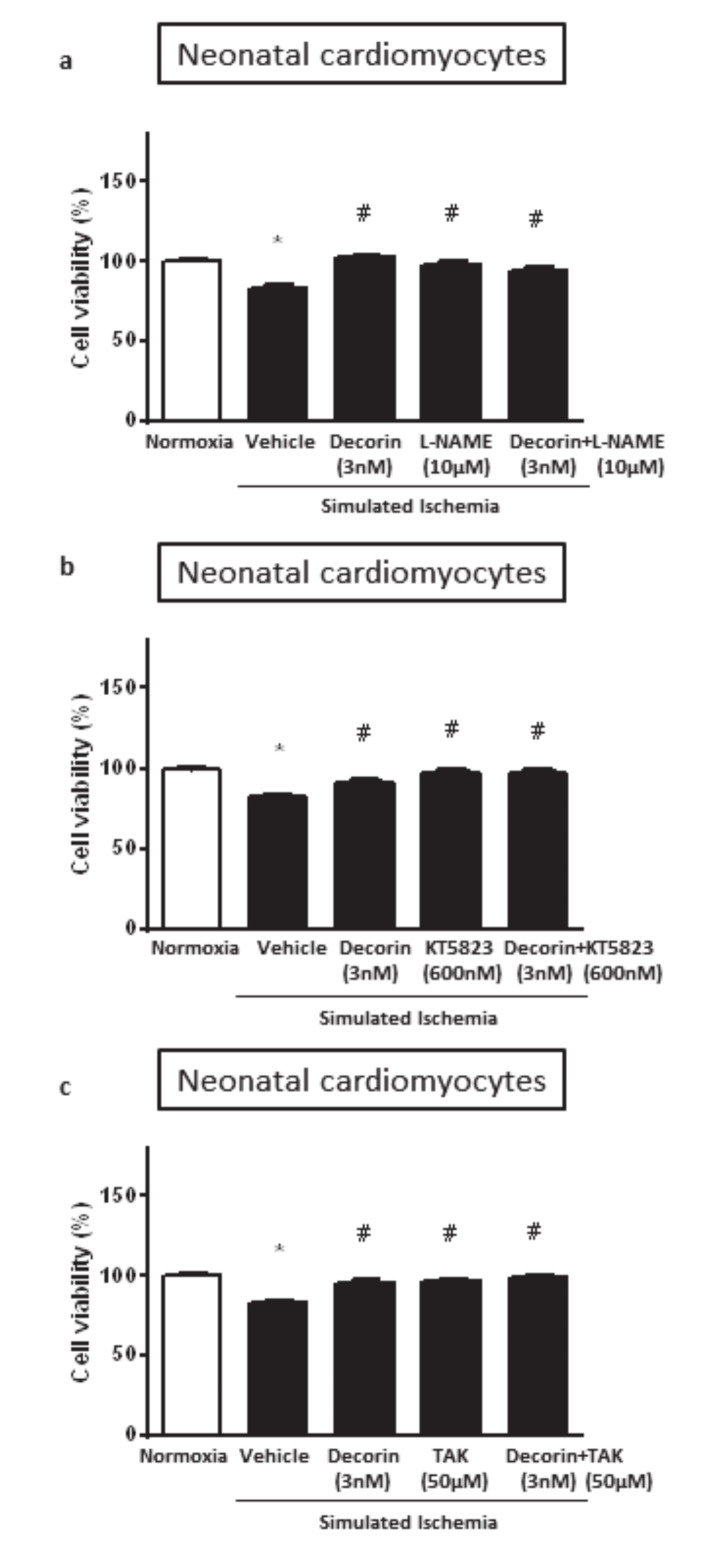
The effect of 3 nM decorin and following inhibitors alone or in combinations: (**a**) 10 µM L-NAME (NOS inhibitor); (**b**) 60 nM KT-5823 (PKG inhibitor); and (**c**) 50 µM TAK-242 (TLR-4 inhibitor); on the cell viability in NRCMs exposed to SI/R. Data are normalized to vehicle-treated Normoxia and presented as mean ± S.E.M. One-Way ANOVA, Dunnett’s multiple comparison test, **p* < 0.05 vs. Normoxia and ^#^*p* < 0.05 vs. Vehicle treated cells (*n* = 5–6).

**Figure 4 molecules-25-03426-f004:**
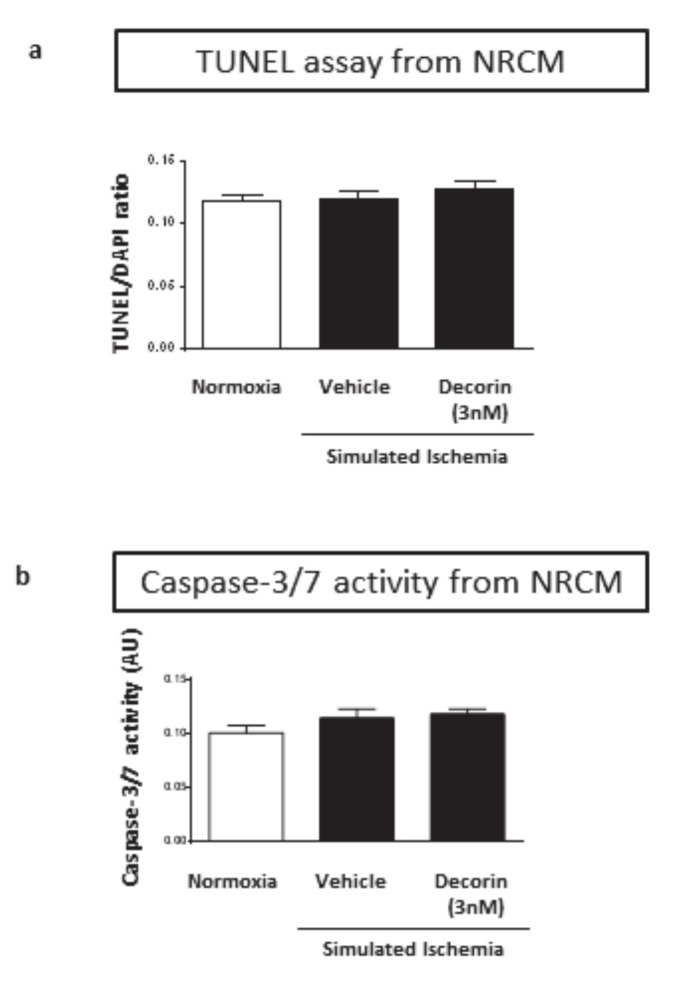
The effect of 3 nM decorin on apoptosis in NRCMs exposed to SI/R. Results of (**a**) TUNEL assay and (**b**) Caspase assay. Data are normalized to vehicle-treated Normoxia and presented as mean ± S.E.M. One-Way ANOVA, Dunnett’s multiple comparison test, **p* < 0.05 vs. normoxia (*n* = 4).

**Figure 5 molecules-25-03426-f005:**
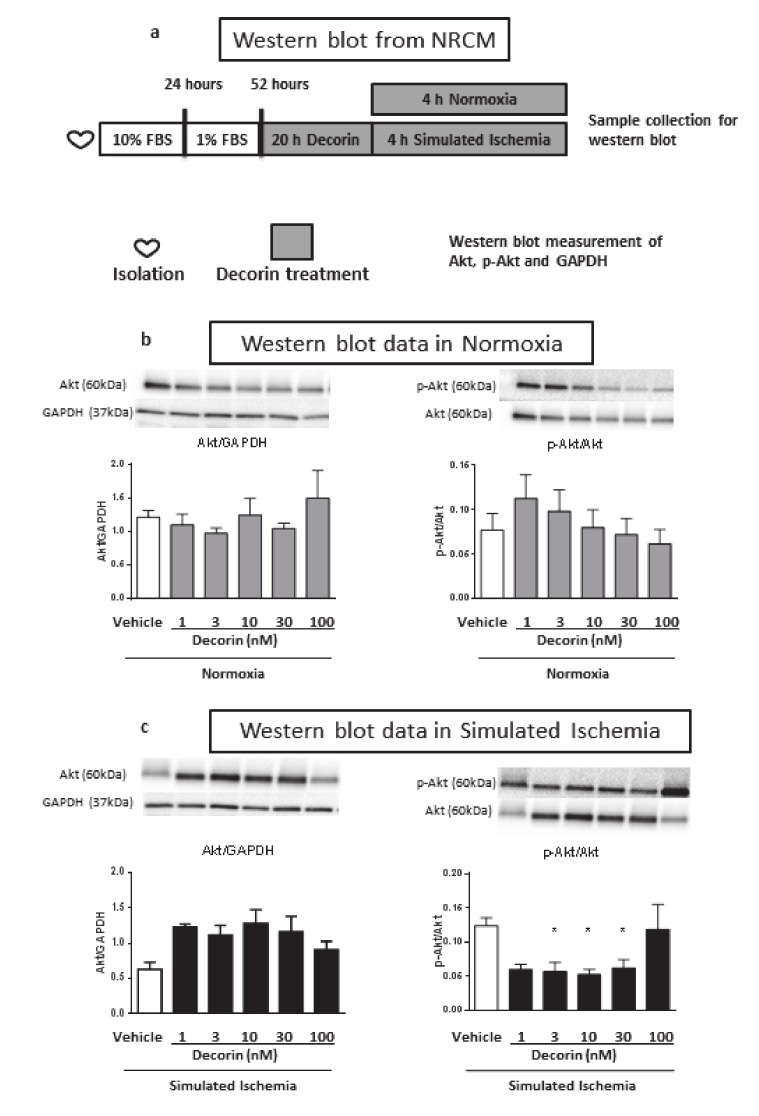
The effect of 1, 3, 10, 30 and 100 nM decorin on Akt, p-Akt protein level in NRCMs exposed to SI/R or Normoxia. (**a**) Experimental protocol of western blot sample collection. (**b**) Akt/GAPDH expression ratio and p-Akt/Akt ratio in Normoxic conditions. (**c**) Akt/GAPDH expression ratio and p-Akt/Akt ratio in SI/R. Representative western blot images are included above quantified column diagram. Data are presented as mean ± S.E.M., One-Way ANOVA, Fisher post hoc test, **p* < 0.05 vs. normoxia (*n* = 2–4).

**Figure 6 molecules-25-03426-f006:**
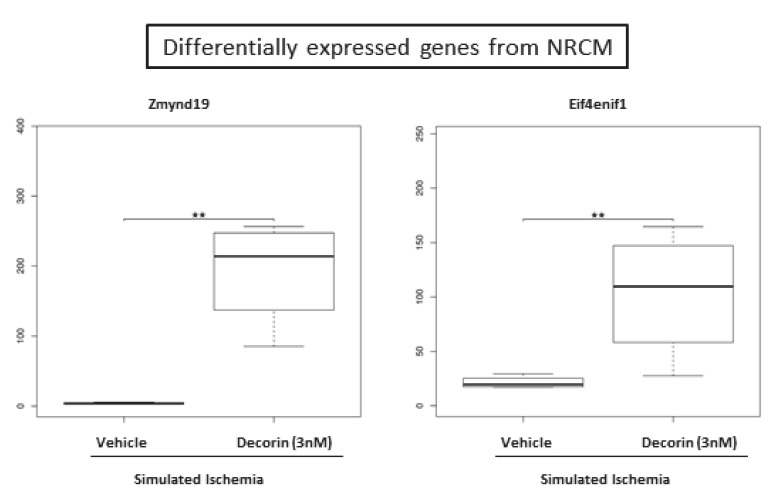
Significantly differentially expressed mRNAs in the decorin-treated NRCM samples compared to the vehicle treated group. p-values and corrected p-values (false-discovery rate according to Benjamini and Hochberg) were calculated by the DESeq2 software package. **denotes corrected *p*-values <0.05.

**Figure 7 molecules-25-03426-f007:**
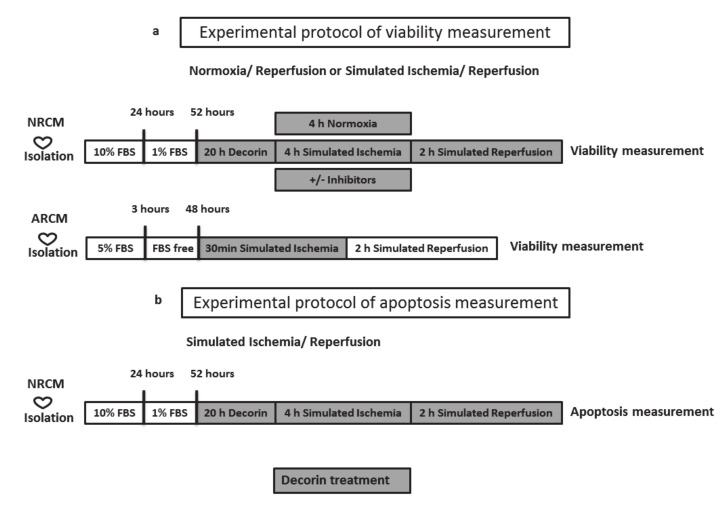
Experimental protocol of (**a**) viability and (**b**) apoptosis measurement. After decorin treatment cell’s viability was measured by calcein assay and apoptosis was measured by TUNEL and caspase assay. NRCM: neonatal rat cardiac myocytes, ARCM: adult rat cardiac myocytes, FBS: fetal bovine serum.

**Table 1 molecules-25-03426-t001:** Gene ontology biological process enrichment analysis of the differentially expressed transcripts upon Decorin treatment. Fold enrichment shows the ratio of the number of the uploaded genes compared to expected number according to reference list. Fold enrichment value over 1 denotes overrepresentation, meanwhile below 1 denotes underrepresentation of the particular process, respectively. Raw *p*-values were obtained by Fisher’s exact test. Corrected *p* values were calculated according to Benjamini-Hochberg correction (false discovery rate).

Gene Ontology (GO) Biological Process Name and Code	Fold Enrichment	Raw *p*-Value	Corrected *p*-Value
Response to oxidative stress (GO:0006979)	2.90	2.21 × 10^−5^	1.78 × 10^−2^
Response to antibiotic (GO:0046677)	2.71	3.83 × 10^−5^	2.95 × 10^−2^
Mitotic cell cycle (GO:0000278)	2.61	6.59 × 10^−5^	4.83 × 10^−2^
Cellular macromolecule metabolic process (GO:0044260)	1.53	4.34 × 10^−6^	5.00 × 10^−3^
Organonitrogen compound metabolic process (GO:1901564)	1.48	9.12 × 10^−6^	9.20 × 10^−3^
Nitrogen compound metabolic process (GO:0006807)	1.48	1.25 × 10^−7^	2.24 × 10^−4^
Cellular metabolic process (GO:0044237)	1.46	3.79 × 10^−8^	8.74 × 10^−5^
Macromolecule metabolic process (GO:0043170)	1.44	5.80 × 10^−6^	6.24 × 10^−3^
Organic substance metabolic process (GO:0071704)	1.42	2.76 × 10^−7^	3.72 × 10^−4^
Primary metabolic process (GO:0044238)	1.40	2.14 × 10^−6^	2.66 × 10^−3^
Metabolic process (GO:0008152)	1.39	2.46 × 10^−7^	3.61 × 10^−4^
Biological_process (GO:0008150)	1.11	1.43 × 10^−5^	1.28 × 10^−2^
Unclassified (UNCLASSIFIED)	54	1.43 × 10^−5^	1.36 × 10^−2^
Sensory perception (GO:0007600)	29	1.73 × 10^−5^	1.47 × 10^−2^
G protein-coupled receptor signaling pathway (GO:0007186)	18	7.61 × 10^−9^	3.07 × 10^−5^
Sensory perception of chemical stimulus (GO:0007606)	10	2.42 × 10^−7^	3.91 × 10^−4^
Sensory perception of smell (GO:0007608)	05	8.43 × 10^−8^	1.70 × 10^−4^
Detection of stimulus involved in sensory perception (GO:0050906)	05	2.61 × 10^−8^	7.03 × 10^−5^
Detection of stimulus (GO:0051606)	04	5.29 × 10^−9^	2.85 × 10^−5^
Detection of chemical stimulus involved in sensory perception of smell (GO:0050911)	<0.01	8.69 × 10^−9^	2.81 × 10^−5^
Detection of chemical stimulus involved in sensory perception (GO:0050907)	<0.01	3.87 × 10^−9^	3.12 × 10^−5^
Detection of chemical stimulus (GO:0009593)	<0.01	2.68 × 10^−9^	4.32 × 10^−5^

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
