# Peer review of "Decorin Protects Cardiac Myocytes against Simulated Ischemia/Reperfusion Injury"

_molecules, 2020, doi:10.3390/molecules25153426_

Round 1

Reviewer 1 Report

By using in vitro culture system, the authors found the protective effect of exogenous Decorin on cardiomyocytes. Recently, Gubbiotti et al. has used in vivo mouse model and reported that Decorin has the cardioprotective effect on cardiomyocytes by regulating autophagy (Gubbiotti et al. Metabolic reprogramming of murine cardiomyocytes during autophagy requires the extracellular nutrient sensor decorin. J Biol Chem. 2018) and Vu et al. has comprehensively summarized the role of Decorin on several signaling pathways in cardiovascular diseases (Vu et al. The role of decorin in cardiovascular diseases: more than just a decoration. Free Radic Res. 2018). Considering these circumstances, I have to require more data to validate and advance science. (1) The authors only use in vitro culture system in the current manuscript. It is difficult for authors to extract solid conclusions only by in vitro experiment expecially in research assessing therapeutic approaches. The authors should use in vivo ischemia-reperfusion model to validate the cardioprotective effect of Decorin. (2) Decorin is generally expressed from cardiac fibroblats in the heart. The authors have to consider the influence of Decorin expressed from cardiac fibroblasts. The authors should perform the cardiomyocyte and cardiac fibroblast co-culture experiment to investigate the function of Decorin from cardiac fibroblasts for example using siRNA or shRNA perturbation on cardiac fibroblasts. (3) The authors showed that Decorin treatment repressed PI3K-Akt signaling, but did not show other possible mechanisms. The authors should conduct unbiased comprehensive analysis such as RNA-seq to reveal the underlying molecular mechanisms, which will strongly support the authors conclusion.

Author Response

Dear Reviewer 1,

Thank you for your revision. For our answers please see the attachment.

Kind regards,
The authors

Reviewer 2 Report

The manuscript submitted by Monika Barteková / Anikó Görbe and co-workers treats about role of decorin, a proteoglycan of extracellular matrix, in protection of cardiac myocytes against stimulated ischemia/reperfusion injury.

The paper is interesting however faces some shortcomings indicated below:

-The Results section should be modified as in the current form it pretty much describes separately every single chart. Each subparagraph should present thematically related set of data.

-It is not mentioned in the methods section how purity of cardiac myocytes was analysed. Any cytostatics were added to cell culture to inhibit possible fibroblasts growth?

-The authors measure viability and proliferation of cells. How the authors define both parameters and what is the double time of population for the cells used to the experiments?

-The Method section misses the details regarding the experiments, e.g what was the number of cells that was used for each assay; how much protein (µg) was loaded onto the gel in WB.

-Thera are number of misspellings, e.g a space between digital and unit should be added; butanedionemonoxime, please replace with correct name of the chemical.

Author Response

Dear Reviewer 2,

Thank you for your revision. For our answers please see the attachment.

Kind regards,
The authors

Round 2

Reviewer 1 Report

The authors have not appropriately responded to my concerns. Since cardiac biology researchers including the present authors seek to understand the molecular mechanisms of the heart, we have to modestly perform in vivo experiments to validate the findings identified in vitro experiments. Papers that are not supported by solid evidence are not necessary for future science. In order to improve the authors manuscript, I suggested several recommendations. but the authors have performed nothing. Unfortunately, I cannot agree with the publication in this manuscript.

Reviewer 2 Report

Dear Authors, Thank you for the answers. The paper was singnificantly improved. One more thing, nitric oxide is a radical and this I suggest to indicate, e.g. NO
